# Sequential production of gametes during meiosis in trypanosomes

Lori Peacock[1,2], Chris Kay[1], Chloe Farren[1], Mick Bailey[2], Mark Carrington [3] & Wendy Gibson [1✉]

Meiosis is a core feature of eukaryotes that occurs in all major groups, including the early diverging excavates. In this group, meiosis and production of haploid gametes have been described in the pathogenic protist, *Trypanosoma brucei*, and mating occurs in the salivary glands of the insect vector, the tsetse fly. Here, we searched for intermediate meiotic stages among trypanosomes from tsetse salivary glands. Many different cell types were recovered, including trypanosomes in Meiosis I and gametes. Significantly, we found trypanosomes containing three nuclei with a 1:2:1 ratio of DNA contents. Some of these cells were undergoing cytokinesis, yielding a mononucleate gamete and a binucleate cell with a nuclear DNA content ratio of 1:2. This cell subsequently produced three more gametes in two further rounds of division. Expression of the cell fusion protein HAP2 (GCS1) was not confined to gametes, but also extended to meiotic intermediates. We propose a model whereby the two nuclei resulting from Meiosis I undergo asynchronous Meiosis II divisions with sequential production of haploid gametes.

[1] School of Biological Sciences University of Bristol, Bristol, UK. [2] Bristol Veterinary School, University of Bristol, Bristol, UK. [3] Department of Biochemistry, University of Cambridge, Cambridge, UK. ✉email: w.gibson@bris.ac.uk

The Excavata is a major eukaryote group that contains diverse deep-branching protist lineages notable for their range of unique features[1–3]. As excavates diverged early in eukaryote evolution, shared eukaryote attributes, such as meiotic sex[4], shed light on the nature of the last common ancestor. Meiosis-specific genes are present in several excavate genomes[5], with experimental evidence for genetic exchange in *Giardia*[6] and the Kinetoplastea[7–11]. While various mechanisms of genetic exchange such as cell fusion may explain the production of hybrids in some of these organisms[12], the pattern of gene inheritance in crosses of *Trypanosoma brucei* indicates that meiosis is involved[13], and cells in Meiosis I and haploid gametes have been found in the salivary glands (SGs) of the tsetse fly vector, where trypanosome mating occurs[14,15].

In *T. brucei*, meiosis starts with a diploid epimastigote cell that sequentially expresses MND1, DMC1 and HOP1—three meiosis-specific proteins key to chromosomal recombination during Meiosis I[5]—and ends with the production of haploid gametes of two types[14,15] (Fig. 1). Each type of gamete has a single haploid nucleus and a kinetoplast (the tightly packaged mitochondrial DNA of trypanosomes) associated with the single flagellum, but in about half the gametes there is an extra kinetoplast[15]. In trypanosomes, the kinetoplast and the basal body (BB) of the flagellum are physically connected by a filamentous structure, the tripartite attachment complex (TAC)[16,17]. The presence of p166, a component protein of the TAC[18], was demonstrated for both kinetoplasts in 2K1N (two kinetoplasts, one nucleus) cells, but no visible flagellum was associated with the extra kinetoplast found in these gametes[15]. The one kinetoplast, one nucleus (1K1N) and 2K1N conformations may represent male and female gametes, though there is no evidence to support this from studying their interactions in vitro[15]. There is currently a gap in knowledge about the process by which the haploid gametes arise from the 4C cell in Meiosis I prophase (meiotic divider, Fig. 1).

To close this gap, we aimed to identify intermediate stages in trypanosome meiosis. This is not straightforward as meiotic dividers and gametes are found in small numbers in the SGs of infected tsetse flies[14,15]. Nonetheless, if these sexual stages are present, it follows that intermediates should also be found. The identity of potential meiotic intermediates could be confirmed if they expressed meiosis-specific proteins, but much the same cellular machinery is used for Meiosis II and mitotic division, so there is a dearth of suitable candidates. On the other hand, there are gamete-specific proteins, such as HAP2 (hapless 2 or generative cell specific1, GCS1), which might serve to identify cells in the later stages of meiotic division. HAP2 is thought to have appeared early in eukaryote evolution and is a key protein involved in the fusion of the membranes of gametes or mating cells in protists, plants and invertebrates[19–26]. HAP2 is homologous to viral fusion proteins; the surface loops of the trimeric protein insert into the membrane of the opposite gamete, causing membrane fusion[25]. HAP2 was thought to be on the surface of only one of the two fusing gametes, e.g. the minus gamete in the single-celled alga, *Chlamydomonas*; the male gamete in the malaria parasite, *Plasmodium* and the plant *Arabidopsis*[19]; however, the ciliate *Tetrahymena* expresses HAP2 on cells of all mating types[22]. In *T. brucei*, there is a *HAP2* homologue on chromosome 10 (Tb927.10.10770[23,26]), but its expression and function are yet to be elucidated in trypanosomes.

Here we used expression of HAP2 as a means to identify gametes and intermediate stages in meiosis in *T. brucei*. The discovery that some intermediates had multiple nuclei with different DNA contents provided an unexpected clue that formed the basis for a model of trypanosome meiosis.

## Results and discussion

**Relative numbers of meiotic dividers, gametes and intermediates.** Meiotic dividers and gametes are most abundant in the tsetse SGs around 21 days post infection, several days before dividing epimastigotes and infective metacyclics come to dominate the population. They occur in a ratio of approximately 1:4, e.g. for strain 1738, there were 48 meiotic dividers and 211 gametes, and for strain J10, 17 meiotic dividers and 72 gametes, an overall total of 19% meiotic dividers and 81% gametes. The two types of gametes (1K1N and 2K1N) are usually produced in approximately equal numbers, e.g. for a total of 405 gametes in strain 1738, there were 199 (49%) 1K1N and 206 (51%) 2K1N gametes, and for strain J10, 32 (44%) 1K1N and 40 (56%) 2K1N gametes; however, there was a predominance of 2K1N gametes in strain TREU 927, where a total of 220 gametes comprised 45 (20%) 1K1N and 175 (80%) 2K1N gametes.

If meiotic dividers and gametes occur together in the SG-derived population, the intermediate stages must also be present, so we carried out a systematic screen of the whole SG-derived trypanosome population, classifying each cell as either a known morphological type (trypomastigote, epimastigote, metacyclic, meiotic divider, gamete) or a potential meiotic intermediate; the latter were defined as trypanosomes with multiple nuclei and/or kinetoplasts, which could not be classified as normal dividing trypomastigotes or epimastigotes[27]. For this experiment, we used two genetically modified clones of strain J10 (J10 *YFP::HOP1 PFR1::YFP* and J10 *YFP::DMC1 PFR1::YFP*), allowing us to unequivocally identify cells in meiosis by the expression of the yellow fluorescent protein (YFP)-tagged meiosis-specific genes *HOP1* and *DMC1*. Among 236 cells examined, there were 17 (7%) meiotic dividers, 72 (31%) gametes and 50 (21%) potential meiotic intermediates (Supplementary Table 1), which were examined further.

Among the potential meiotic intermediates, we expected to find trypanosomes with two kinetoplasts and nuclei (2K2N), resulting from division of the 4C nucleus of the meiotic cell (Fig. 1), but also present were large numbers of 3K2N, 3K3N and 4K2N cells, besides smaller numbers of other categories of multinucleate/multikinetoplast cells (Table 1); this result was confirmed by analysis of three clones from two other *Trypanosoma brucei* strains, adding a further category of 3K1N cells (Table 1). It is usually straightforward to identify trypanosome kinetoplasts by their small size relative to the nucleus after DNA staining and their position adjacent to the proximal end of the flagellum, but here the relative sizes of the kinetoplasts in a single cell were variable, as previously observed for gametes[15]; moreover, the association with a flagellum was often obscure and sometimes

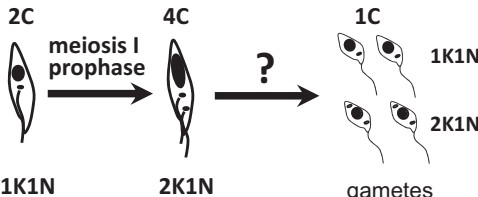

**Fig. 1 Model of meiosis in *Trypanosoma brucei*.** On the left, an epimastigote [kinetoplast (black dot) anterior to nucleus (black oval)] enters meiosis. Meiotic dividers expressing the meiosis-specific proteins MND1, DMC1 or HOP1 have the morphology shown for the 4C trypanosome: a large posterior nucleus, two kinetoplasts, and two flagella[14]. Haploid (1C) gametes are of two types: one kinetoplast, one nucleus (1K1N) and 2K1N[15] and it is assumed that four are produced from each trypanosome that enters meiosis. The question mark signifies the current gap in knowledge about intermediate stages in meiosis.

the nucleus and kinetoplast overlapped. Hence, some kinetoplast identifications may have been erroneous and we therefore needed additional criteria to identify organelles.

**Identification of organelles**. The BB can be recognized using a monoclonal antibody specific for tyrosinated α-tubulin (YL1/2[28]), which accumulates at the base of the flagellum[29]; tyrosinated α-tubulin is also a marker for newly formed microtubules, since they become detyrosinated with time[30,31]. We therefore used YL1/2 immunofluorescence to verify whether each kinetoplast was associated with a BB (Fig. 2). While this was the case for both kinetoplasts in dividing procyclics from culture, as reported by

others[29–31], usually only the posterior BB was stained in meiotic dividers (9 of 10) and dividing epimastigotes (4 of 6) isolated from tsetse SG (Fig. 2b, c); the lack of YL1/2 signal implies the absence of a pool of tyrosinated tubulin associated with the anterior BB and old flagellum. In 13 1K1N gametes, the single kinetoplast was invariably associated with a BB as expected (Fig. 2e), but this was the case in only 3 of the 17 2K1N gametes (Fig. 2f), although we previously demonstrated that the TAC protein p166 was associated with both kinetoplasts in 2K1N cells[15]. Thus, YL1/2 did not provide a reliable means to identify the BB, and indirectly the kinetoplast, in trypanosomes with known morphologies, raising doubts about the interpretation of YL1/2 signal in putative meiotic intermediates such as those shown in Fig. 2g–k. The numbers of BBs and Ks did not always tally (e.g. Fig. 2g–i, k), though in some cells the intensity of fluorescence at the posterior could have obscured a faint BB signal (e.g. Fig. 2i). While two prominent BBs were visible in intermediates with two flagella (Fig. 2g, h), this was not always the case (Fig. 2i), and conversely, two strongly stained BBs were present in trypanosomes with a single flagellum (Fig. 2j).

As an alternative means to identify kinetoplasts, their mitochondrial location was verified by dual fluorescence of 4,6-diamidino-2-phenylindole (DAPI) staining the kinetoplast DNA and green fluorescent protein (GFP) targeted to the mitochondrial lumen via the N-terminal import signal peptide of frataxin Tb927.3.1000 (1738 *Mito GFP*, Fig. 3a–e). All kinetoplasts in 1K, 2K, 3K and 4K cells co-localized with frataxin fluorescence, implying that all kinetoplasts were inside the mitochondrion, though this might be inevitable in cells with a highly ramified mitochondrion (e.g. Fig. 3c).

As the large size of some kinetoplasts made them difficult to distinguish from nuclei by DAPI staining alone (e.g. Figs. 2e and 3e), nuclei were visualized using a histone 2B fusion protein

**Table 1 Morphology of predominant types of meiotic intermediates among salivary gland-derived trypanosomes of T. b. brucei.**

| Cell type (ordered by no. of nuclei) | J10 PFR1::YFP YFP::HOP1 and PFR1::YFP YFP::DMC1 | 1738 Mito GFP | 1738 H2B::GFP PFR1::YFP | TREU 927 HAP2::YFP |
|---|---|---|---|---|
| | N = 50 | N = 138 | N = 214 | N = 351 |
| 3K1N | 0 (0%) | 20 (15%) | 5 (2%) | 67 (19%) |
| 2K2N | 8 (16%) | 32 (23%) | 57 (27%) | 54 (15%) |
| 3K2N | 11 (22%) | 21 (15%) | 50 (23%) | 73 (21%) |
| 4K2N | 12 (24%) | 14 (10%) | 33 (15%) | 46 (13%) |
| 3K3N | 6 (12%) | 10 (7%) | 18 (8%) | 11 (3%) |

Four data sets are listed, comprising data from salivary gland-derived trypanosomes of five different recombinant clones from three strains of *T. b. brucei* (J10, 1738, TREU 927). Overall, a total of 56 cytospins of pooled, dissected salivary glands were examined, derived from ~950 tsetse flies (J10, 12 cytospins, ~200 flies; 1738 *Mito GFP*, 5 cytospins, ~100 flies; 1738 *H2B::GFP*, 31 cytospins, ~500 flies; TREU 927, 8 cytospins, ~150 flies). Only the most numerous cell types (≥10%) in one or more trypanosome clones are shown; percentages relate to the total number of cells categorized (*N*).

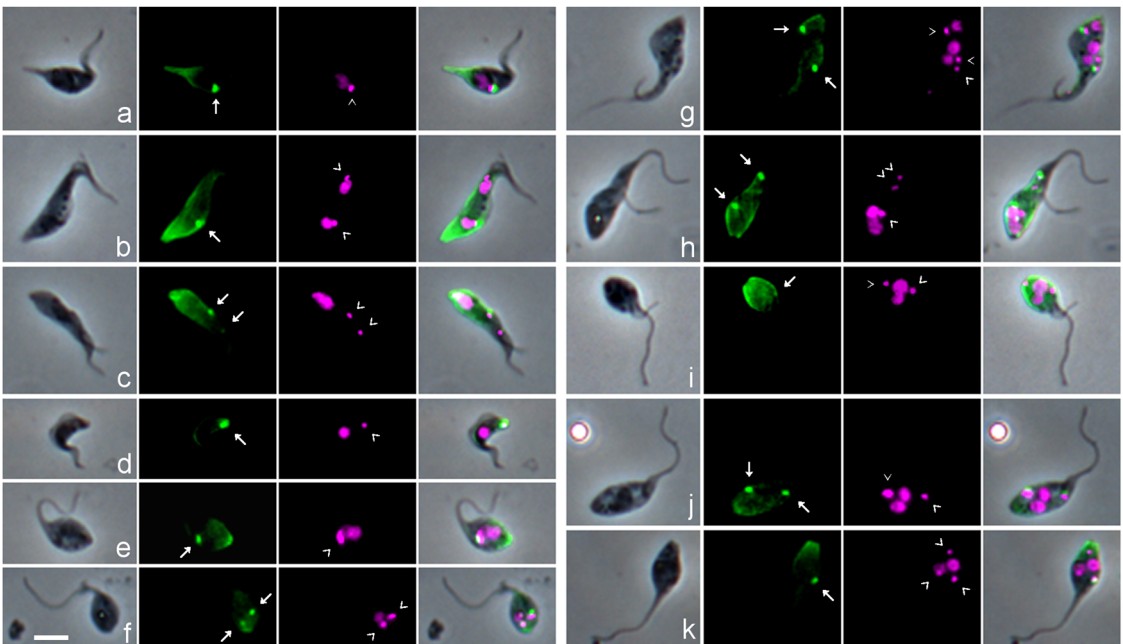

**Fig. 2 Identification of the basal body using YL1/2 antibody against tyrosinated α-tubulin.** *T. b. brucei* 1738 wild-type trypanosomes derived from tsetse salivary glands were fixed and stained with YL1/2 antibody against tyrosinated α-tubulin visualized with FITC secondary antibody; the basal bodies (BBs) shown as bright green fluorescent dots, with additional staining of the plus ends of newly formed microtubules at the posterior of the cell. L to R: phase contrast, YL1/2 (green FITC fluorescence), DAPI (false coloured magenta fluorescence), merge all. BBs are indicated by white arrows (YL1/2 images) and kinetoplasts (K) by arrowheads (DAPI images). **a** 1K1N epimastigote with prominent posterior nozzle; **b** 2K2N dividing epimastigote; **c** 2K1N meiotic divider, with just visible anterior BB; **d** 1K1N metacyclic; **e** 1K1N gamete; **f** 2K1N gamete. **g–k** various meiotic intermediates with multiple K and/or N: **g** 3K3N cell with two flagella; **h** 3K2N cell with two flagella; **i** 2K1N cell with two flagella; **j** 2K2N, single flagellum; **k** 3K2N, single flagellum. Scale bar = 5 μm.

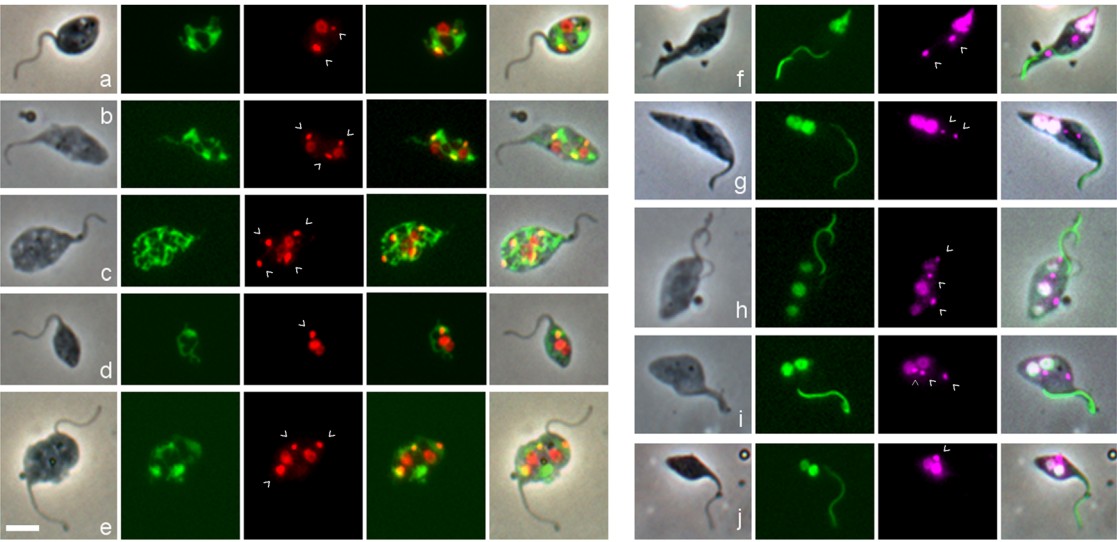

**Fig. 3 Visualization of the mitochondrion and nucleus in gametes and potential meiotic intermediates. a–e** In *T. b. brucei* 1738 *Mito GFP*, GFP is targeted to the mitochondrial lumen using the 24 amino acid import signal of frataxin. L to R: phase contrast, GFP, DAPI (red), merge GFP and DAPI, merge all. DAPI fluorescence is coloured red to visualize yellow (i.e. dual red and green) fluorescence of the kinetoplast within the mitochondrion in the merged images. Kinetoplasts are indicated by arrowheads. **a** 2K1N gamete; **b** 3K2N intermediate; **c** 4K2N intermediate; **d** 1K2N intermediate; **e** 3K2N intermediate with two flagella. **f–j** *T. b. brucei* 1738 *H2B::GFP YFP::PFR*, fusion proteins visualize the nucleus and the paraflagellar rod (PFR) in the flagellum respectively. L to R: phase contrast, GFP, DAPI (magenta), merge. **f** 2K1N meiotic divider with very large posterior nucleus; the paler PFR belongs to the new flagellum; **g** 2K2N intermediate; **h** 3K3N intermediate; **i** 3K2N intermediate; **j** 1K2N intermediate. Scale bar = 5 µm.

(1738 *H2B::GFP*, Fig. 3f–j). H2B is a very stable nuclear marker, as it is part of the nucleosome core and remains bound to nuclear DNA throughout the cell cycle, except for a brief period during DNA replication. H2B mRNA levels are maximum during S phase of the cell cycle[32,33], congruent with the coupling of DNA and histone synthesis in eukaryotes. Hence, we assume that all the H2B synthesized during Meiosis I subsequently partitions equally among the haploid nuclei of the gametes. This is supported by observation: all 48 meiotic dividers and the majority of gametes (204 of 211, 97%) expressed H2B::GFP in the nucleus in SG-derived trypanosomes of *T. b. brucei* 1738 *H2B::GFP PFR1::YFP*. As there is a single copy of *H2B::GFP* per diploid nucleus (Supplementary Fig. 1), only half the gametes will inherit a copy and be capable of synthesizing H2B::GFP de novo.

An intriguing observation was that some of the meiotic intermediates appeared to have nuclei of different sizes, judging by the intensity of both the H2B and DAPI fluorescence (e.g. Fig. 3h, j), whereas a trypanosome undergoing binary fission typically has nuclei of the same size. To quantify this phenomenon more precisely, we measured H2B nuclear fluorescence in multinucleate cells.

**Quantitation of nuclear DNA content using *H2B::GFP* expression.** We measured the relative fluorescence of H2B::GFP in each nucleus of individual binucleate (2N) and trinucleate (3N) cells, recording cell morphology at the same time; examples of cells with nuclei of markedly different fluorescence intensities are shown in Fig. 4a–j. A total of 232 cells were measured, comprising 190 2N and 42 3N trypanosomes. Among the 190 2N cells, there were 34 normal dividing trypomastigotes or epimastigotes, assumed to have nuclei of equal size; the remaining 156 2N cells had been categorized as meiotic intermediates and either had nuclei of approximately equal fluorescence intensity or nuclei that appeared unequal. Correspondingly the per cell ratios of nuclear fluorescence intensities for the 190 2N cells showed a bimodal distribution, with peaks centred on 0.5 and 1.0 (Fig. 5a). Since fluorescence intensity is proportional to the amount of H2B::GFP,

which in turn is proportional to the amount of nuclear DNA, the nuclear DNA contents of some 2N cells are in a 1:2 ratio and the simplest interpretation is that these are 1C and 2C nuclei (though in principle the 1:2 ratio also applies to 2C:4C nuclei or any multiple of this).

For each 3N cell, one nucleus was visibly more fluorescent than the other two, which were of approximately equal intensity. Figure 5b shows the ratio of the nuclear fluorescence intensity of each small nucleus compared to the large one in each 3N cell; the unimodal distribution is centred on 0.5, indicating a 1:2 ratio between each small nucleus and the larger one in the same 3N cell, i.e. one 2C and two 1C nuclei.

The presence of 1C nuclei is direct and unequivocal evidence that these 2N and 3N cells are intermediates in meiosis. It is unlikely that they result from fusion of a gamete with another cell, as intraclonal mating is rare in trypanosomes[34,35]. For example, when SG-derived trypanosomes of red and green fluorescent clones of *T. b. brucei* were mixed in vitro, yellow fluorescent cells were observed routinely in mixtures of different strains but seldom in intraclonal mixtures[36]. Here in contrast, the putative meiotic intermediates were relatively abundant in SG-derived populations, e.g. 2N and 3N trypanosomes with unequal nuclei together comprised just over half (142, 53%) of the 267 cells identified as putative meiotic intermediates in the SG-derived population of 1738 *H2B::GFP PFR1::YFP*.

The order of the 1C and 2C nuclei from posterior to anterior of the 3N cell varied (Supplementary Table 2), suggesting that either of the 2C nuclei divides, and as the two 1C nuclei pull apart on a longitudinal spindle, one moves past the remaining 2C nucleus, so that the final, and most abundant, conformation is 1C 2C 1C (Fig. 4f). Similarly, the order of the 1C and 2C nuclei in 2N cells varied: among 87 2N cells, the 1C nucleus was posterior in 43 cells (50%) (Fig. 4a, b), anterior in 34 (39%) (Fig. 4c) and adjacent to the 2C nucleus in 10 (11%) (Fig. 4d, e).

In 13 3N cells, cytokinesis was almost complete, separating cells with one or two nuclei (Fig. 4g–j); in seven cases, the daughter cell cleaving from the posterior was either a 2K1N or

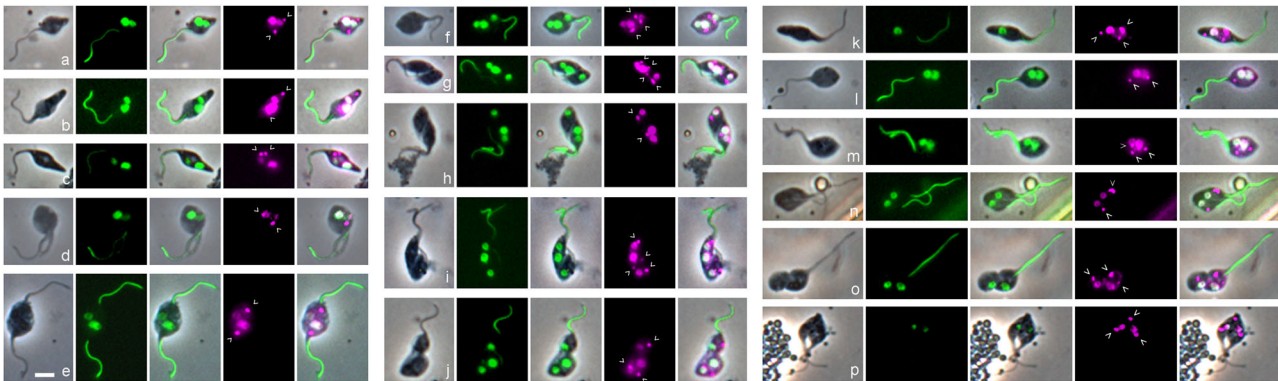

**Fig. 4 Morphology of meiotic intermediates from tsetse salivary glands.** Salivary gland-derived trypanosomes of 1738 *H2B::GFP PFR1::YFP* were fixed and stained with DAPI. L to R: phase contrast, GFP, merge, DAPI (magenta), merge all. **a–j** show individual trypanosomes with nuclei of different fluorescence intensities. **a–e** show trypanosomes with two nuclei of different fluorescence intensities; **a**, **b** show a single 2K2N cell where the smaller nucleus is posterior, while in **c** the smaller nucleus is anterior; **d**, **e** show dividing 2K2N cells with two flagella. **f–j** trypanosomes with three nuclei of different fluorescence intensities; **f** 3K3N cell, where the central nucleus appears larger than the other two; **g–i** show a 1K1N daughter cell with a small nucleus cleaving from a 2N trypanosome with two nuclei of different fluorescence intensities, while in **j** the 2K2N cell at the posterior is cleaving from a 1K1N gamete at the anterior, and the 2K2N "daughter" cell has nuclei of unequal fluorescence. **k–p** show the final stages of meiosis. **k** 3K1N cell with a single long flagellum; **l–p** gamete-like trypanosomes with two nuclei of equal intensity; two flagella are evident in cells **m**, **n**, **p**, but only a single flagellum is visible in cells **l**, **o**, so the daughter flagellum is either running closely parallel with the original or has not formed; **o** 1K1N gamete cleaving from the posterior of a 2K1N gamete, while in **p** cytokinesis is almost complete, yielding one 1K1N and one 2K1N gamete. Kinetoplasts are indicated by arrowheads. Scale bar = 5 μm.

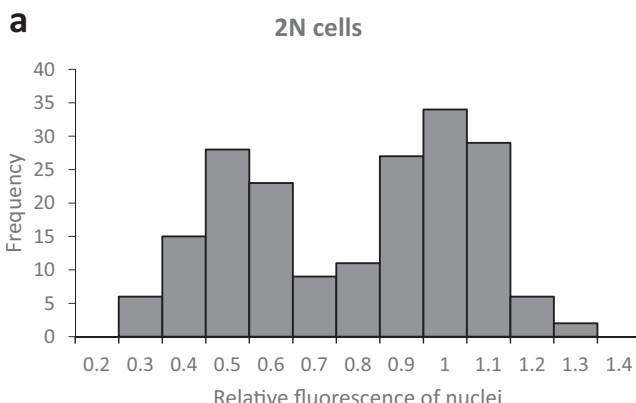

**a**

**2N cells**

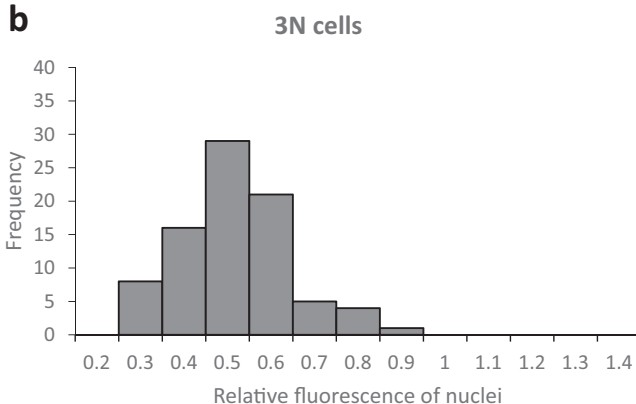

**b**

**3N cells**

**Fig. 5 Ratio of fluorescence intensities of nuclei within individual 2N and 3N cells.** The total fluorescence intensity of each individual nucleus within a cell was measured and the ratio between nuclei in the same cell was determined. For 2N cells (**a**, *n* = 190), the ratio of fluorescence intensities of both nuclei in the same cell is shown; for 3N cells (**b**, *n* = 42), the ratio of fluorescence intensities of each of the two small nuclei was compared with that of the large nucleus. The figures on the *x*-axis are the upper bound of each bin; bin size 0.1.

1K1N haploid gamete (Fig. 4g–i), but in six cases, the "daughter" cell cleaving from the posterior of a gamete was a 2N trypanosome with nuclei of different fluorescence intensities (Fig. 4j). In normal trypanosome cell division, the daughter cell forms at the posterior of the mother cell and inherits the newly constructed flagellum[30,31], so there is a clear distinction between mother and daughter cell. Here, identification of the daughter cell was based on its posterior position only.

**Expression of HAP2 in gametes and potential meiotic intermediates.** We investigated whether expression of the gamete fusion protein HAP2 would identify gametes and their precursors. Procyclic cells were transfected with a *HAP2::YFP* fusion construct designed to integrate into the *HAP2* locus with control of expression via the endogenous promoter, 5′ and 3′ untranslated regions (UTRs). Transfected *T. brucei* clones were then cycled through tsetse flies to track expression of the HAP2 fusion protein by fluorescence microscopy in different life cycle stages. For all four *T. brucei* strains tested (TREU 927, STIB 247, J10, 1738), expression was detected only among trypanosome life cycle stages in the fly SGs, where meiosis occurs and gametes are found[14,15]; no expression was detected in midgut procyclics or proventricular trypanosomes.

We analysed SG-derived trypanosomes from TREU 927 in depth (Table 2). Of the 1478 cells examined, 656 (44.3%) expressed HAP2. The vast majority of these cells (649 of 656, 99%) were either gametes (185 of 656, 28%, including both 1K1N and 2K1N) or potential meiotic intermediates (464 of 656, 71%), as previously defined. Unexpectedly, 2 of the 12 meiotic dividers expressed HAP2, though these cells are in an early stage of meiosis (prophase I), but only negligible numbers of HAP2 expressors were found among epimastigotes, metacyclics and other trypomastigotes. While the majority of gametes expressed HAP2, 16% (35 of 220 gametes) were negative, comprising 10 1K1N and 25 2K1N gametes (Table 2). It is possible (though unlikely given the small proportion) that these non-expressors were female gametes, as only male gametes express HAP2 in other organisms; however, HAP2 is present on both mating partners in ciliates[22].

**Table 2 Expression of *HAP2::YFP* in salivary gland-derived trypanosomes of TREU 927.**

| Cell type | Number of HAP2-expressing cells (%) | Number of non-expressing cells (%) | Total |
|---|---|---|---|
| Gametes (1K1N, 2K1N) | 185 (84.1%) | 35 (15.9%) | 220 |
| 1K1N gametes | 35 (77.8%) | 10 (22.2%) | 45 |
| 2K1N gametes | 150 (85.7%) | 25 (14.3%) | 175 |
| Potential meiotic intermediates | 464 (90.4%) | 49 (9.6%) | 513 |
| Epimastigotes (1K1N) | 4 (1.9%) | 206 (98.1%) | 210 |
| Meiotic dividers (2K1N) | 2 (16.7%) | 10 (83.3%) | 12 |
| Metacyclics | 1 (0.1%) | 388 (99.9%) | 389 |
| Trypomastigotes | 0 (0.0%) | 96 (100%) | 96 |
| Dividing trypomastigotes | 0 (0.0%) | 38 (100%) | 38 |
| Total | 656 (44.3%) | 822 (55.6%) | 1478 |

Flies dissected 18–23 days post-infection. All cells expressing *HAP2::YFP* were imaged and samples of mostly non-expressing cell types (epimastigotes, meiotic dividers, metacyclics, trypomastigotes) were also counted to estimate the proportion of expressors.

In some gametes, there were noticeable accumulations of the HAP2 fusion protein, sometimes at the cell posterior (Fig. 6a), but other cells showed several spots of bright fluorescence and/or generalized fluorescence throughout the cytoplasm (Fig. 6b); this might suggest storage of HAP2 in cytoplasmic vesicles before deployment[21]. There was no discernible difference between the cellular distribution of HAP2 expression in 1K1N and 2K1N gametes. Similar patterns of expression were observed among the potential meiotic intermediates (Fig. 6c–e). When these HAP2-expressing meiotic intermediates were categorized by the numbers of nuclei and kinetoplasts, the largest categories, ordered by size, were: 3K2N (21%), 3K1N (19%), 2K2N (15%), and 4K2N (13%), the same cell types that featured prominently among potential meiotic intermediates in other *T. b. brucei* strains (Table 1). Expression of HAP2 strengthens the case that these cells are the precursors to gametes, i.e. meiotic intermediates that appear late in the process.

**Model of meiosis**. To summarize, we have three separate lines of evidence that point to particular cell morphologies as potential meiotic intermediates: unusual cell conformation with multiple nuclei and/or kinetoplasts; two or three nuclei with different DNA contents in the same cell; expression of HAP2. We now need to link these intermediates into a logical series of events in order to formulate a model of meiosis in *T. brucei*.

Both the meiotic divider (intermediate A, Fig. 7) and the 3N cell with one 2C and two 1C nuclei (intermediates C1 and C2, Fig. 7) have a total DNA content of 4C and therefore the 3N cell should follow on from the meiotic divider, presumably with a stage in between in which the 4C nucleus undergoes a conventional reduction division yielding two 2C nuclei, without cytokinesis (intermediate B, Fig. 7). Intermediate B has two 2C nuclei of equal size and could be confused with mitotically dividing epimastigotes or trypomastigotes. However, in these cells the nuclei and kinetoplasts appear in a particular order from the posterior of the cell; thus dividing epimastigotes are NKNK, dividing trypomastigotes are KNKN and epimastigote/trypomastigote dividers are NKKN[27]. Hence, 2N cells that do not conform to one of these arrangements, particularly because they have 3 or 4K (49/59, 83%, Supplementary Table 3), are potential candidates for intermediate B; some examples are shown in Fig. 3g, i. We infer that kinetoplast replication occurs in intermediate B, though no more than two flagella were evident, and this is consistent with

the multiple K observed in intermediates C1 and C2, where the majority (67%, 35 of 52) had 3 or 4 K (Supplementary Table 3). Intermediates A, B and C1 were indistinguishable in terms of cell size parameters (Fig. 8a, b), implying that the changes affecting the nuclei and kinetoplasts do not alter cytoplasmic volume or cell shape.

When C1 divides (intermediate C2, Fig. 7), one of the 1C nuclei together with one or more kinetoplasts, pass into the first gamete produced; examples are shown in Figs. 2g and 4g–j. The dimensions of this gamete concur with those of free gametes (Fig. 8c). The remaining cell is a 2N intermediate with a total DNA content of 3C, as it has one 2C and one 1C nucleus (intermediate D1, Fig. 7); examples are shown in Fig. 4a–c. The number of K was variable, though note that some K might be in the process of replication: among the 74 D1 cells observed, 20 (27%) were 1K2N, 37 (50%) were 2K2N, 13 (18%) were 3K2N and the remaining 4 (5%) were 4K2N (Supplementary Table 3). Production of a second 1C gamete from D1 would require the construction of a new flagellum (intermediate D2, Fig. 7), presumably necessitating K and BB replication, though it is also feasible that growth of the new flagellum initiates from one of the existing BBs in cells with multiple Ks. Though the examples shown in Fig. 4d, e have two flagella, it is uncertain how such cells complete cytokinesis, as only three examples of intermediate D3 were found, none with very good morphology. The dimensions of D1 and D2 cells occupy a wide range, partly overlapping with that of C2-Mother, the predecessor of D1 (Fig. 8d), suggesting changes in shape during transition from C2 to D2 (Fig. 7).

The remaining cell with one 2C nucleus (intermediate E1, Fig. 7) is also difficult to identify with confidence, as it resembles a gamete, because of its relatively long flagellum and small body. However, gametes are either 1K1N or 2K1N, so gamete-like 3K1N cells are likely to be intermediate E1 (e.g. Fig. 4k), and the appearance of a new flagellum defines intermediate E2 (Fig. 7). Of the 30 intermediate E1/E2 cells identified, 13 (43%) were 2K1N and 17 (57%) were 3K1N (Supplementary Table 3); examples are shown in Figs. 4k, l and 6c.

The following stage of division into two 1C gametes (intermediates F1 and F2, Fig. 7) is easier to identify, as there are two nuclei of equal fluorescence (assumed to be 1C) in a characteristic gamete-like cell with one or two flagella; of the 36 such cells, 4 (11%) were 1K2N, 14 (39%) were 2K2N and 18 (50%) were 3K2N (Supplementary Table 3); examples are shown in Fig. 4l–p. Presumably construction of a new flagellum requires replication of the K-BB complex, as for the transition from intermediate D1 to D2. Depending on the number of kinetoplasts present, these dividing cells may produce two 1K1N gametes or one 1K1N and one 2K1N gamete in an asymmetric division, as shown in Fig. 4o, p; this inconsistency may account for the fact that equal proportions of 1K1N and 2K1N gametes are not always achieved.

The sequence of events shown in Fig. 7 roughly agrees with the relative proportions of HAP2 expressors (Table 1), assuming that HAP2 is maximally expressed by gametes and their immediate precursors. Thus, 3K2N (21%) and 3K1N (19%) cells were prominent among HAP2 expressors, comprising intermediates F1/F2 and E1/E2, respectively, while intermediates C1/C2 (3K3N) were less well represented (3%) (Table 1). Although HAP2 expression is restricted to gametes or mating cells in other species, here HAP2 expression in intermediate stages may reflect the fact that gametes are being formed sequentially rather than just at the end of meiosis. HAP2 is unlikely to be in its final conformation/ location in these intermediate stages to avoid the possibility of premature membrane fusion; we envisage that only mating-competent gametes should express HAP2 in its active form[25,26]. However, in experimental crosses polyploid hybrids are often

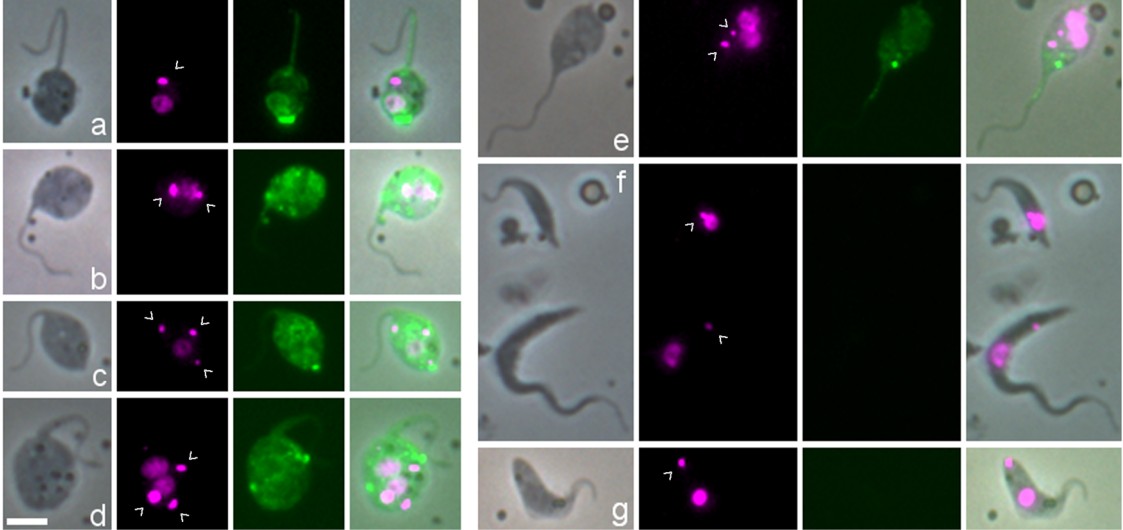

**Fig. 6 Expression of _HAP2::YFP_ in SG-derived _T. b. brucei_ TREU 927.** L to R: phase contrast, DAPI (magenta), YFP, merge. **a–e** Examples of individual trypanosomes expressing _HAP2::YFP_. Gametes: **a** 1K1N; **b** 2K1N. Potential meiotic intermediates: **c** 3K1N; **d** 3K2N; **e** 2K2N. Examples of non-expressing cells: **f** top, epimastigote; bottom, trypomastigote; **g** metacyclic. Kinetoplasts are indicated by arrowheads. Scale bar = 5 μm.

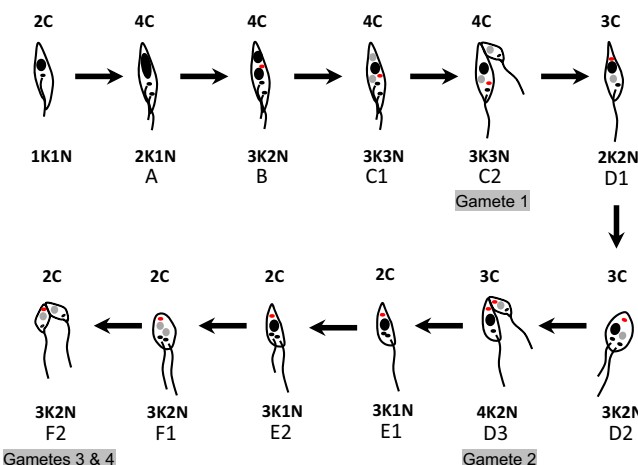

**Fig. 7 Model of meiosis based on observed intermediates.** Nuclei are shown in black (4C or 2C) or grey (1C). Kinetoplasts associated with a flagellum are shown in black, while extra kinetoplasts not associated with a flagellum are shown in red. For clarity, flagella are drawn as separate lines, though two distinct flagella were not always evident in imaged cells. The total DNA content of each trypanosome is indicated above the cell, while the relative numbers of kinetoplasts (K) and nuclei (N) are shown below, but note that the numbers of K varied in stages B–F2 (Supplementary Table 3). Intermediates A–F are described in the text.

produced along with diploid progeny[37–39], suggesting that fusion not only occurs between gametes but also between gametes and intermediates. This might be facilitated by premature expression of HAP2 in 2C cells such as E1/E2 since these cells are high HAP2 expressors.

## Conclusion

Of the potential meiotic intermediates we have identified here, the 2N and 3N trypanosomes with 1C and 2C nuclei were key to developing a model of the likely sequence of events during meiosis. Although the model we have derived does not fit with the symmetrical nuclear divisions described in textbook accounts of eukaryote meiosis, asymmetrical meiotic division is not unusual among eukaryotes, a case in point being human oogenesis, where polar bodies are sequentially discarded, leaving only a single haploid egg[40]. However, the asynchronous division of nuclei suggested here occurs within the shared cytoplasm and would require a special control mechanism, perhaps only feasible in cells with a closed mitosis and meiosis.

The model differs significantly from trypanosome mitosis, which follows a strictly ordered sequence of events, with kinetoplast and BB replication at the start, allowing the new flagellum to be constructed during nuclear DNA synthesis, leading to cytokinesis after all structures have been replicated[16,30,31]. Meiosis follows this pattern up to a point, but then kinetoplast replication and construction of new flagella become, of necessity, uncoupled from nuclear DNA synthesis, because the 4C nucleus needs only to divide. Kinetoplast replication could be linked to nuclear division, but a fixed pattern of events was not obvious, which is difficult to reconcile with the tight control of the mitotic cell cycle in trypanosomes[30,31].

Although meiosis is thought to be an ancestral eukaryotic feature[4,5], trypanosomes represent an ancient, deeply branching lineage, with the insignia of this divergence displayed in unique organelles, such as the kinetoplast and glycosome. It is therefore not unreasonable to suggest that the evolution of meiosis in trypanosomes may have taken a different path after divergence, constrained by the need to coordinate reduction division with replication of the flagellum in a rigidly structured, asymmetric cell.

## Experimental procedures

**Trypanosomes.** The following tsetse-transmissible strains of _T. b. brucei_ were used: _T. b. brucei_ J10 (MCRO/ZM/73/J10 CLONE 1), 1738 (MOVS/KE/70/EATRO 1738), TREU 927 (GPAL/KE/70/EATRO 1534), and STIB 247 (MALC/TZ/71/STIB 247). Procyclic form trypanosomes were grown in Cunningham's medium (CM)[41] supplemented with 15% v/v heat-inactivated foetal calf serum, 5 μg/ml hemin and 10 μg/ml gentamycin at 27 °C. Tsetse were infected with trypanosomes and dissected essentially as described previously[14,15].

**Transfection.** Procyclic trypanosomes were transfected by electroporation, antibiotic selected and cloned as previously described[14].

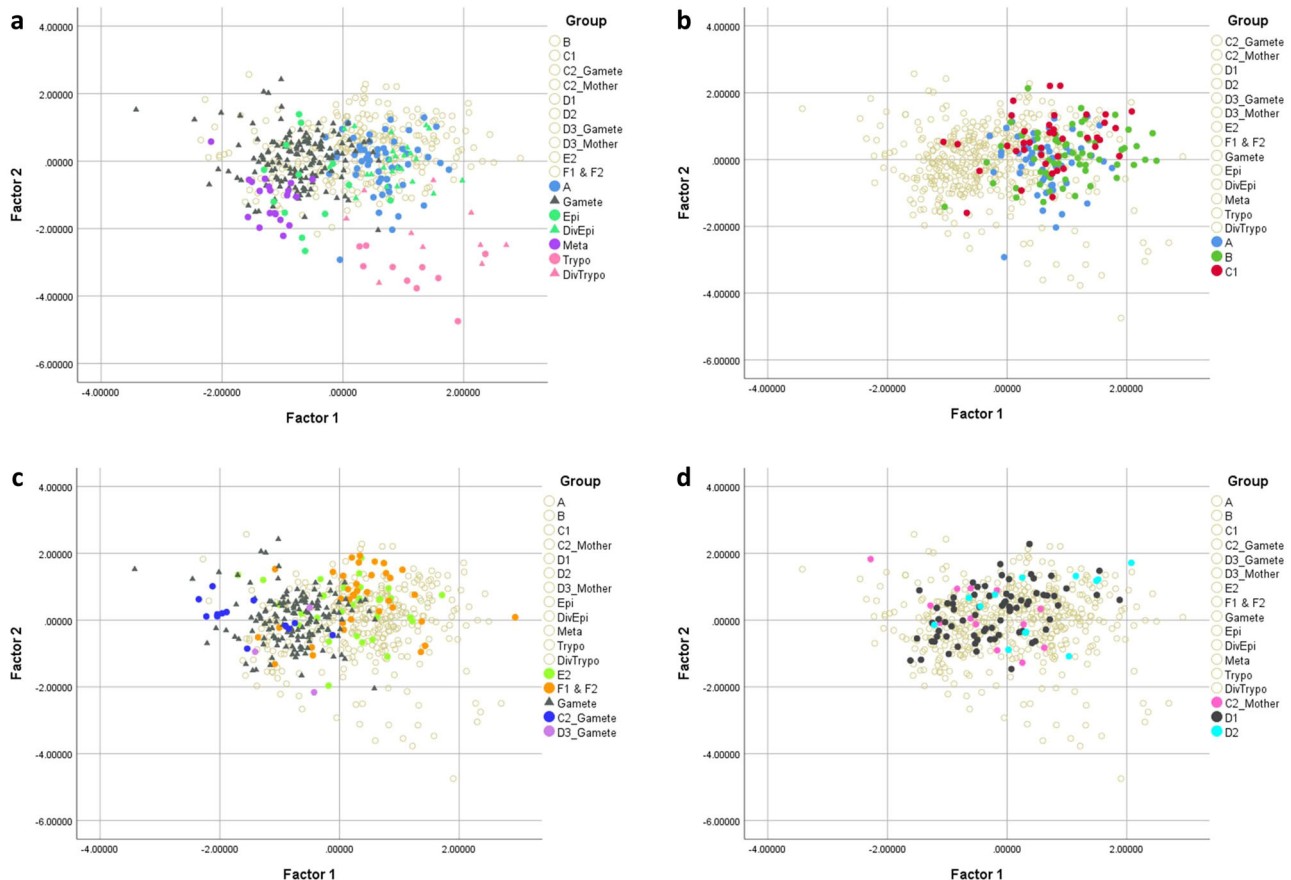

**Fig. 8 Principal components analysis of cell morphology. a** Comparison of meiotic dividers (intermediate A) with gametes and other cell types from the salivary glands (metacyclics, epimastigotes, dividing epimastigotes, trypomastigotes, and dividing trypomastigotes). **b** Comparison of cell types A, B, and C1. **c** Comparison of gametes produced in intermediates C2 and D3 with free gametes and dividing gametes (intermediates E2, F1, and F2). **d** Comparison of intermediate D1 with its progenitor (C2 mother) and D2.

The following fluorescently tagged genes were used to visualize cellular structures: *PFR1::YFP* incorporated into the paraflagellar rod of the flagellum[14]; *H2B::GFP* incorporated into the nucleus (*Tb927.10.10460*) via a plasmid construct targeted for integration into the tubulin locus and designed to express *H2B::GFP* with the *H2B* 3′ UTR. In addition, the mitochondrion was visualized by inserting the N-terminal 24 amino acid mitochondrial targeting sequence of frataxin (*Tb927.3.1000*; nucleotides 1–72) upstream of *GFP* in a plasmid construct targeted for integration into the tubulin locus. The following fluorescently tagged genes were used as stage-specific markers: *YFP::DMC1* and *YFP::HOP1* expressed in the nucleus during meiosis[14]; *HAP2::YFP*, *HAP2* (*GCS1*, *Tb927.10.10770*) fused in frame with *YFP* followed by the *HAP2* 3′ UTR downstream and targeted for integration into the endogenous locus. This plasmid construct and two transfected trypanosome lines (TREU 927, STIB 247) were kindly supplied by Eva Gluenz and Keith Gull (University of Oxford, UK).

**Microscopy.** Fly SGs were dissected directly into CM to recover trypanosomes spilling from the broken ends of the glands. Aliquots of the supernatant were fixed for 30 min in 2% w/v paraformaldehyde (PFA) at room temperature (RT) and then spread on microscope slides using a cytospin. Slides were stained with DAPI in VECTASHIELD mounting medium (Vector Laboratories) and viewed using a DMRB microscope (Leica) equipped with a Retiga Exi camera (QImaging) and Volocity software (PerkinElmer). The whole area of the cytospin was scanned

systematically from top to bottom, capturing GFP/YFP, DAPI and phase-contrast images of each trypanosome.

**Immunofluorescence.** Fly SG were dissected directly into CM as above and the supernatant fixed for 30 min in 4% w/v PFA at RT, followed by permeabilisation with 1:200 v/v Triton in distilled water for 7 min at 4 °C. Cells were pelleted by centrifugation and washed twice in phosphate-buffered saline (PBS), then allowed to settle on a round glass coverslip. Cells were incubated with 2% w/v bovine serum albumin (BSA) for 30 min at RT, followed by a further 30-min incubation with the primary antibody (YL1/2, Merck Life Science) diluted 1:100 in 2% BSA. Following removal of the primary antibody by three PBS washes, a secondary anti-rat IgG fluorescein isothiocyanate-conjugated antibody (Life Technologies) diluted 1:200 in 2% BSA was added and incubated for 30 min at RT. After three PBS washes, cells were stained with DAPI in VECTASHIELD mounting medium and imaged as above.

**Image analysis.** Digital images were collated and analysed using ImageJ (http://rsb.info.nih.gov/ij). The previously described macro for measurement of DNA content in DAPI-stained cells[15] was used to measure GFP fluorescence in *H2B::GFP* nuclei by changing the target image (green instead of blue channel). Mensural data were collected from 554 SG-derived trypanosomes of 1738 *H2B::GFP PFR1::YFP*, comprising total cell length

including free flagellum, cell width at the widest point, cell body length not including free flagellum and cell body area (Supplementary Data 1).

**Statistics and reproducibility.** Mensural data were subjected to principal component (PC) analysis using the statistical package IBM SPSS Statistics for Windows, Version 26. Two uncorrelated factors, PC1 and PC2, accounted for 84% of the variance. Extracted scores for PC1 and PC2 were plotted for each trypanosome.

*Reproducibility.* Multiple tsetse transmission experiments were performed in order to obtain sufficient SG trypanosomes for analysis. Most experiments were done with *T. b. brucei* 1738, either wild type or various genetically modified clones, but experiments were also repeated with other strains of *T. b. brucei*.

*Sample sizes.* In total, 56 cytospin preparations of trypanosomes from pooled SG dissected from ~950 tsetse flies were examined by phase contrast and fluorescence microscopy. Morphology of 753 trypanosomes was examined and 554 cells were subject to mensural analysis; 1478 trypanosomes were examined for HAP2 expression. Additional transmission experiments were done with 1738 wild type to recover trypanosomes from tsetse organs for immunofluorescence using YL1/2 antibody.

**Reporting summary**. Further information on research design is available in the Nature Research Reporting Summary linked to this article.

## Data availability
All data generated or analysed during this study are included in this published article and its supplementary information files.

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

## Acknowledgements

This work was funded by BBSRC grant BB/R010188/1 to W.G. and M.B. M.C. is a Wellcome Trust Investigator (217138/Z/19/Z). We thank Storm McReady-Fallon for transfection of the *H2B::GFP* nuclear constructs, Isabel Roditi for advice on immuno-fluorescence, Eva Gluenz for kindly supplying the *HAP2::YFP* construct and transfected trypanosome cell lines, Keith Gull for early discussions raising awareness of HAP2 in trypanosomes and two anonymous reviewers for constructive criticism of earlier versions of this paper. We gratefully acknowledge the generous supply of tsetse pupae from the International Atomic Energy Agency, Vienna.

## Author contributions

W.G. and M.B. applied for funding. L.P. carried out the bulk of the experimental work with contributions from C.K., C.F., M.C. and W.G. M.B. and L.P. carried out data analysis. C.K., C.F. and M.C. provided materials. W.G. drafted the manuscript and all authors read and approved the final version.

## Competing interests

The authors declare no competing interests.
