## [Transparent Peer Review File · Communications Biology]

Reviewers' comments:

Reviewer #1 (Remarks to the Author):

The authors have previously described *Trypanosoma brucei* cells undergoing Meiosis I and identified haploid gametes. Here, to investigate the missing parts of trypanosome meiosis, they searched for intermediate stages inside the tsetse salivary glands. From the cell types recovered, they deduced that haploid gametes are produced sequentially, as the two 2C nuclei resulting from Meiosis I seemed to undergo asynchronous divisions. They also report that the expression of the fusion protein HAP2 was not confined to gametes, but also extended to meiotic intermediates.

The authors used a fine and elegant combination of image analyses of cells endogenously expressing a tagged-HAP2 version and of their genetic material. Although this manuscript is based on the careful manual scrutinization of an impressive number of individual cells recovered from a significant amount of fastidious work with experimentally infected arthropods, the conclusions may appear limited to a model that still remains very hypothetical.

Of the potential meiotic intermediate cells, the 2N and 3N trypanosomes with 1C and 2C nuclei were indeed key to developing a model of the possible sequence of events during meiosis (the ratios of fluorescence intensities of nuclei within individual 2N and 3N cells described in Fig. 5 are very nice!). However, a significant proportion of observations do not fit the model. For instance, how do the authors explain that 15% of the gametes (and up to 1/5 1K1N gametes) do not express HAP2? How do they integrate the presence of cells with more than 4K in their model? Why is the morphology of most intermediate cells so unusually rounded? As stated in the discussion, this model does not fit with the symmetrical nuclear divisions described in textbook for eukaryote meiosis and their observations are also difficult to reconcile with the tight control of the mitotic cell cycle in trypanosomes. If this model is plausible and tempting, it only partially fits the data integration presented in the manuscript and this should be more clearly explained and carefully presented and discussed. Proposing some possible further experiments to be performed in this direction would also be a plus. If applicable, using an unbiased approach, such as PCA, and with a larger set of bio-morphological parameters would have been more convincing to discriminate each relevant intermediate steps of meiosis.

Whereas the use of HAP2 as a marker for meiotic intermediate is original and convincing, the first approach relying on kinetoplast analysis is weaker as it is presented in the manuscript. The part « Relative numbers of meiotic dividers, gametes and intermediates » is not easy to follow. Multiple switches between the main manuscript and the supplementary data doesn't help the reader to keep the main narrative flow. Some of these supplementary data, especially those already published, are probably not necessary, whereas some key aspects defining observation criteria should be placed in the main text. In this part, the main text is sometimes difficult to follow. One would have expected the absolute numbers and proportions of all stages identified according to well-defined morphological traits and molecular markers to be presented at first. Then, the definition of normal Vs. large kinetoplasts is not clearly stated in the main text and the authors sometimes use without distinctions the kinetoplast size and fluorescence intensity, that are different parameters which could vary according to the experimental procedure (e.g. cell fixation method or focal plan). For this kind of detailed and subtle observations, the use of a confocal microscope would have been more adapted. For instance, the standard deviations of the fluorescence intensity measurements in Supplementary Table 1 are high and do not permit to draw any conclusion.

Another concern is the general morphology of most intermediate cells presented in this manuscript. L117, the authors comment that, since most aberrant forms are easily dismissed as products of faulty division, large numbers need to be considered as intermediates in a defined process. What are the exact criteria used to easily dismiss products of faulty divisions without any other molecular markers and knowing that meiotic intermediates may also appear as odd cells? In Fig2, the morphology and organization of most of the cells (at the exception of f and j) looks odd. For instance, the cell in picture

e is possibly dying and that in picture h presents a strange flagellar labelling. Do all the cells at each given possible meiotic intermediate step look the same (size, general morphology, organelle distribution, flagellum length...)? Could the authors provide different pictures of intermediate cells for each configuration in the same panel? Can the authors exclude an experimental alteration of the parasite morphology after SG dissection, pooling of SG extracts and cytospin treatment? Did they confirm their observations by directly monitoring parasites through the tissues of intact glands thanks to the fluorescent markers?

In the abstract, the authors highlight the fact that, in contrast to other eukaryotes, the trypanosome flagellum is retained throughout meiosis, with new flagella constructed as each new cell is produced. However, this interesting aspect is only briefly discussed in the manuscript. Are the free kinetoplasts already associated to tripartite attachment complex, basal body and / or intraflagellar transport proteins? It would have been nice to confirm the presence of these flagellum-associated structures or precursors by performing simple IFAs. The mechanisms and timing for the construction of the new flagella associated to the free kinetoplasts in meiotic intermediates is very intriguing!

Minor comments:

- L104: Is Sup Table 3 appropriately referred to here?
- L111: Supplementary Data 1 shows the total fluorescence intensities of kinetoplasts not the relative sizes of these organelles.
- Sup Table 4 should be placed in the main manuscript.
- In most experiments, the number of metacyclic parasites appears quite low. How do you explain this?
- Fig. 7: The cell shapes in the cartoon do not correspond to what is shown in the picture panels. Most of the meiotic intermediates are actually much rounded. This could represent an advantage for the intracellular reorganization by offering more space and a higher flexibility to the multiple organelle movements occurring during meiosis.

Reviewer #2 (Remarks to the Author):

This paper describes the uncommon progression of meiosis in *Trypanosoma brucei* and will be of interest for readers from the meiosis and eukaryote evolution fields. The authors identified *Trypanosoma* cell types in fly salivary glands which are not of the previously classified stages and express Hap2, which is found in gametes (and possibly also in pre-gametic stages), and thus define them as cells in meiosis I. Judging from the number, size and (indirect) measurements of DNA contents of nuclei, they categorize them as the products of first and second meiotic divisions. Based on the presence of cells with putative single meiosis I plus two meiosis II products, they propose that the second meiotic division of the products of the first meiotic division occur asynchronously (but in a controlled manner). This phased meiosis would result in the sequential production of gametes. If true, this would be a remarkable deviation from the canonical meiotic process and strong evidence for the early divergence of meiosis in excavates. Alternatively, uncoordinated second meiotic division could reflect the unreliability of a "primitive" meiosis. While there is a slight possibility that meiosis of *Trypanosoma* is error prone and that cells with unexpected numbers of nuclei produce infertile cull, compelling evidence for sequential meiosis comes from the (mostly) "fitting" numbers and sizes of nuclei in the gametes. The paper is clearly written, although I found references to separate supplemental materials and supplemental tables somewhat confusing. Further confirmation for the nature of intermediate meiosis stages could come from the study of knockouts of MND1, DMC1 or HOP1, which probably would perturb nuclei numbers and sizes in affected cells. But the paper in its present form is a good starting point for further elucidating details of the unusual meiosis of Trypanosomes.

Specific comments:

- 1) The title is somewhat unspecific. The singular is not warranted because meiosis consists of two divisions, on the other hand, not both are phased. To be precise, it would have to read "Phased second meiotic division ...". As this may not encompass the real oddity of Trypanosoma meiosis, "Phased meiosis in ..." may be an alternative.
- 2) L 53: Explain abbreviation symbol on first occurrence: "The one kinetoplast-one nucleus (1K1N) and 2K1N configurations"
- 3) Figure 1: Meiosis I encompasses all steps from premeiotic (or as it is now called: meiotic) S phase to telophase I. Therefore it is incorrect to designate the step from 1K1N to 2K1N as "meiosis I", as there occur only premeiotic S and meiotic prophase. It should be better called "meiotic prophase".
- 4) L 106: Better: "... corresponding to symmetrical first meiotic division of the 4C cell ..."
- 5) Fig. 2i: One of the three kinetoplasts seems to overlap with a nucleus, so it is difficult to see. In general it would be good to indicate Ks and Ns with arrows where the situation is unclear.
- 6) L 161: "meiotic dividers (= cells in Meiosis I)" Move this explanation up to L 56-57
- 7) L 171: "There was no discernible difference between HAP2 expression in 1K1N and 2K1N gametes." I don't quite understand: If there was a difference (1:2.5) in 1K1N and 2K1N among the non-expressors (line 164), then conversely, there must be the reciprocal difference among the expressors.
- 8) In Fig. 4 j, a 2K2N daughter cell (with a presumptive 2C and 1C nucleus) is cleaving from a 1K1N GAMETE. Would the daughter cell continue with a second meiosis and produce three gametes? N.B.: Can a cell that produces a daughter cell be called a gamete? (Same in line 207).
- 9) L 261-262: This wording is confusing: "... the 4C nucleus of the meiotic divider undergoes a phased reduction division in contrast to the equal divisions of conventional meiosis." As you explain, the first meiotic division (4C->2x 2C) is normal (L 294). The first meiotic division is conventionally called the reductional division, thus "phased reduction division" is misleading. It is the second, the equational, division, which is phased, as it initially takes place in only one 2C nucleus (2x 2C -> 2C + C + C).
- 10) L 367-369: The terms "symmetrical" and "asymmetrical divisions" should be avoided. They are rather used in the sense of equal or unequal distribution of chromosomes (e.g., sex chromosomes, B-chromosomes) or equal/unequal sized products (e.g., egg and polar bodies). Better stay with "asynchronous second divisions".
- 11) L 495: Correct spelling of the author is Hörandl
- 12) It would help the reader to see stages shown in Figs 2, 4, and 6 linked to the classes A-C described in Fig. 7, maybe by referring to corresponding panels in Figs 2, 4, 6 in the chapter "Model of meiosis"

Response to Reviewers

We thank the reviewers for their constructive comments on the MS and very much appreciate the two complementary perspectives on the work, one from a trypanosome cell biologist and one from an expert on meiosis. We have extensively revised the MS, adding new data and figures, in light of the reviewers' comments and below is a point by point response.

Reviewer #1 (Remarks to the Author):

The authors have previously described *Trypanosoma brucei* cells undergoing Meiosis I and identified haploid gametes. Here, to investigate the missing parts of trypanosome meiosis, they searched for intermediate stages inside the tsetse salivary glands. From the cell types recovered, they deduced that haploid gametes are produced sequentially, as the two 2C nuclei resulting from Meiosis I seemed to undergo asynchronous divisions. They also report that the expression of the fusion protein HAP2 was not confined to gametes, but also extended to meiotic intermediates.

The authors used a fine and elegant combination of image analyses of cells endogenously expressing a tagged-HAP2 version and of their genetic material. Although this manuscript is based on the careful manual scrutinization of an impressive number of individual cells recovered from a significant amount of fastidious work with experimentally infected arthropods, the conclusions may appear limited to a model that still remains very hypothetical.

Of the potential meiotic intermediate cells, the 2N and 3N trypanosomes with 1C and 2C nuclei were indeed key to developing a model of the possible sequence of events during meiosis (the ratios of fluorescence intensities of nuclei within individual 2N and 3N cells described in Fig. 5 are very nice!). However, a significant proportion of observations do not fit the model. For instance, how do the authors explain that 15% of the gametes (and up to 1/5 1K1N gametes) do not express HAP2? How do they integrate the presence of cells with more than 4K in their model? Why is the morphology of most intermediate cells so unusually rounded? As stated in the discussion, this model does not fit with the symmetrical nuclear divisions described in textbook for eukaryote meiosis and their observations are also difficult to reconcile with the tight control of the mitotic cell cycle in trypanosomes. If this model is plausible and tempting, it only partially fits the data integration presented in the manuscript and this should be more clearly explained and carefully presented and discussed. Proposing some possible further experiments to be performed in this direction would also be a plus. If applicable, using an unbiased approach, such as PCA, and with a larger set of bio-morphological parameters would have been more convincing to discriminate each relevant intermediate steps of meiosis.

We now provide mensural data and PCA plots to support the identification of intermediates (new Figure 8). Cells with more than 4K were observed, but very infrequently in comparison to cells with 1-4K. This, together with the uncertainty of unequivocally identifying kinetoplasts, we think justifies omitting cells with >4K from the model.

Whereas the use of HAP2 as a marker for meiotic intermediate is original and convincing,

the first approach relying on kinetoplast analysis is weaker as it is presented in the manuscript.

We take the point, but have retained the initial screen of morphology of SG-derived cells as a “no assumptions” preliminary approach.

The part « Relative numbers of meiotic dividers, gametes and intermediates » is not easy to follow. Multiple switches between the main manuscript and the supplementary data doesn't help the reader to keep the main narrative flow. Some of these supplementary data, especially those already published, are probably not necessary, whereas some key aspects defining observation criteria should be placed in the main text.

Apologies for this. In the revised MS, we've removed as many references to Supplementary material as possible, instead incorporating the relevant information into the text where possible to improve the narrative flow.

In this part, the main text is sometimes difficult to follow. One would have expected the absolute numbers and proportions of all stages identified according to well-defined morphological traits and molecular markers to be presented at first. Then, the definition of normal Vs. large kinetoplasts is not clearly stated in the main text and the authors sometimes use without distinctions the kinetoplast size and fluorescence intensity, that are different parameters which could vary according to the experimental procedure (e.g. cell fixation method or focal plan). For this kind of detailed and subtle observations, the use of a confocal microscope would have been more adapted. For instance, the standard deviations of the fluorescence intensity measurements in Supplementary Table 1 are high and do not permit to draw any conclusion.

One of the intrinsic problems is that we're attempting to recognise stages in meiotic division, but in reality we are observing cells undergoing a dynamic process, without synchronization at the start. Thus, we only have a snapshot of each cell at a particular point in the process, rather than a sequential series of images from a single cell. For example, we found a variable number of K in the 2N and 3N cells with unequal nuclei, and different positioning of the nuclei relative to the cell posterior. We interpreted these differences in morphology in terms of a dynamic process rather than as individual, static stages. Confocal provides high image quality by eliminating all out-of-focus signal, but here it was important to capture total fluorescence for example to quantitate GFP fluorescence from cell nuclei and visualize multiple kinetoplasts lying in slightly different focal planes.

Another concern is the general morphology of most intermediate cells presented in this manuscript. L117, the authors comment that, since most aberrant forms are easily dismissed as products of faulty division, large numbers need to be considered as intermediates in a defined process. What are the exact criteria used to easily dismiss products of faulty divisions without any other molecular markers and knowing that meiotic intermediates may also appear as odd cells?

The text has been replaced here (lines 127-133, revised MS). What we meant to convey was that you often find odd-shaped cells, maybe damaged during isolation and fixing. Each individual occurrence has little significance by itself, but large numbers of the same odd-shaped cell imply that this cell type has significance.

In Fig2, the morphology and organization of most of the cells (at the exception of f and j) looks odd. For instance, the cell in picture e is possibly dying and that in picture h presents a strange flagellar labelling. Do all the cells at each given possible meiotic intermediate step look the same (size, general morphology, organelle distribution, flagellum length...)? Could the authors provide different pictures of intermediate cells for each configuration in the same panel? Can the authors exclude an experimental alteration of the parasite morphology after SG dissection, pooling of SG extracts and cytospin treatment? Did they confirm their observations by directly monitoring parasites through the tissues of intact glands thanks to the fluorescent markers?

We have used these same protocols extensively for studying salivary gland trypanosomes and recover known cell types such as metacyclics and epimastigotes reliably with good morphology. In the case of new cell types we described previously, such as trypanosomes undergoing meiosis and gametes, we also observed these as live cells directly in the salivary glands or in the saliva spill out. These prior observations give us full confidence that the cell morphologies observed here – atypical though they may be – are real and not caused by preparation artifacts.

In the abstract, the authors highlight the fact that, in contrast to other eukaryotes, the trypanosome flagellum is retained throughout meiosis, with new flagella constructed as each new cell is produced. However, this interesting aspect is only briefly discussed in the manuscript. Are the free kinetoplasts already associated to tripartite attachment complex, basal body and / or intraflagellar transport proteins? It would have been nice to confirm the presence of these flagellum-associated structures or precursors by performing simple IFAs. The mechanisms and timing for the construction of the new flagella associated to the free kinetoplasts in meiotic intermediates is very intriguing!

Yes indeed what happens to the flagellum is interesting! We first reported the extra kinetoplast without a flagellum in gametes and in that paper showed that YFP-tagged p166, a component protein of the TAC, localized to both kinetoplasts, indicating that a basal body is associated with each kinetoplast, though the flagellum is absent. Following the reviewer's suggestion, we tried IFA using the YL1/2 antibody that recognises tyrosinated tubulin in order to identify the basal body. We got some interesting results, now included as new Figure 2 and additional results paragraph (lines 149-168, revised MS) , but it still wasn't able to confirm association of kinetoplast and basal body in all cases.

Minor comments:

- L104: Is Sup Table 3 appropriately referred to here?

Supplementary tables have been removed where possible and numbering revised.

- L111: Supplementary Data 1 shows the total fluorescence intensities of kinetoplasts not the relative sizes of these organelles.

Yes, total fluorescence intensity as a measure of amount of DNA and a proxy for size. This supplementary file was removed, as no longer needed after revision.

- Sup Table 4 should be placed in the main manuscript.

This HAP2 data is now Table 2 in MS.

- In most experiments, the number of metacyclic parasites appears quite low. How do you explain this?

This is because we use an early timepoint (~21 days post infective feed) in trypanosome colonisation of the salivary glands, when meiotic forms and gametes are most abundant but

few metacyclics have yet developed. This has been clarified in the text (lines 101-2, revised MS).

- Fig. 7: The cell shapes in the cartoon do not correspond to what is shown in the picture panels. Most of the meiotic intermediates are actually much rounded. This could represent an advantage for the intracellular reorganization by offering more space and a higher flexibility to the multiple organelle movements occurring during meiosis.

This figure has been redrawn to better represent the real images.

Reviewer #2 (Remarks to the Author):

This paper describes the uncommon progression of meiosis in *Trypanosoma brucei* and will be of interest for readers from the meiosis and eukaryote evolution fields.

The authors identified *Trypanosoma* cell types in fly salivary glands which are not of the previously classified stages and express Hap2, which is found in gametes (and possibly also in pre-gametic stages), and thus define them as cells in meiosis I. Judging from the number, size and (indirect) measurements of DNA contents of nuclei, they categorize them as the products of first and second meiotic divisions. Based on the presence of cells with putative single meiosis I plus two meiosis II products, they propose that the second meiotic division of the products of the first meiotic division occur asynchronously (but in a controlled manner). This phased meiosis would result in the sequential production of gametes. If true, this would be a remarkable deviation from the canonical meiotic process and strong evidence for the early divergence of meiosis in excavates. Alternatively, uncoordinated second meiotic division could reflect the unreliability of a "primitive" meiosis. While there is a slight possibility that meiosis of *Trypanosoma* is error prone and that cells with unexpected numbers of nuclei produce infertile cull, compelling evidence for sequential meiosis comes from the (mostly) "fitting" numbers and sizes of nuclei in the gametes.

The paper is clearly written, although I found references to separate supplemental materials and supplemental tables somewhat confusing.

Apologies for this. In the revised MS, we've removed as many references to Supplementary material as possible, instead incorporating the relevant information into the text where possible.

Further confirmation for the nature of intermediate meiosis stages could come from the study of knockouts of MND1, DMC1 or HOP1, which probably would perturb nuclei numbers and sizes in affected cells. But the paper in its present form is a good starting point for further elucidating details of the unusual meiosis of Trypanosomes.

We have contemplated knockout of meiosis-specific genes to disrupt meiosis. In other systems, this does not block meiosis, but leads to errors in chromosome recombination and segregation and nuclear division – as the reviewer suggests, perturbation in number and size of nuclei. With the complexity of cell types already seen, we feel that correct interpretation of the results might be a struggle! A trypanosomatid with a simpler life cycle might be a more fruitful experimental model, but a suitable organism and a suite of molecular tools is necessary before this is possible.

Specific comments:

1) The title is somewhat unspecific. The singular is not warranted because meiosis consists of two divisions, on the other hand, not both are phased. To be precise, it would have to read "Phased second meiotic division ...". As this may not encompass the real oddity of *Trypanosoma* meiosis, "Phased meiosis in ..." may be an alternative.

Thanks for the suggestion – title revised: “Sequential production of gametes during meiosis in trypanosomes”

2) L 53: Explain abbreviation symbol on first occurrence: "The one kinetoplast-one nucleus (1K1N) and 2K1N configurations"

OK done (line 60 revised MS).

3) Figure 1: Meiosis I encompasses all steps from premeiotic (or as it is now called: meiotic) S phase to telophase I. Therefore it is incorrect to designate the step from 1K1N to 2K1N as "meiosis I", as there occur only premeiotic S and meiotic prophase. It should be better called "meiotic prophase".

Thanks – this has been changed in the revised Figure 1.

4) L 106: Better: "... corresponding to symmetrical first meiotic division of the 4C cell ..."

Thanks – text revised (line 123 revised MS, “resulting from division of the 4C nucleus of the meiotic cell”).

5) Fig. 2i: One of the three kinetoplasts seems to overlap with a nucleus, so it is difficult to see. In general it would be good to indicate Ks and Ns with arrows where the situation is unclear.

Overlapping K and N occurs quite often, as the K is sometimes close to the N. Arrows have been added to all cell image figures to indicate Ks.

6) L 161: "meiotic dividers (= cells in Meiosis I)" Move this explanation up to L 56-57

Thanks – text revised in both places (line 63 “the 4C cell in Meiosis I prophase (meiotic divider, **Figure 1**)”).

7) L 171: "There was no discernible difference between HAP2 expression in 1K1N and 2K1N gametes." I don't quite understand: If there was a difference (1:2.5) in 1K1N and 2K1N among the non-expressors (line 164), then conversely, there must be the reciprocal difference among the expressors.

This referred to the cellular distribution of HAP2 expression rather than on/off – clarified in the text (lines 326-8, revised MS “There was no discernible difference between the cellular distribution of HAP2 expression in 1K1N and 2K1N gametes.”).

8) In Fig. 4 j, a 2K2N daughter cell (with a presumptive 2C and 1C nucleus) is cleaving from a 1K1N GAMETE. Would the daughter cell continue with a second meiosis and produce three gametes? N.B.: Can a cell that produces a daughter cell be called a gamete? (Same in line 207).

As the reviewer notes, this is anomalous, but this configuration was observed in nearly half the dividing cells like this, an example being Fig 4j. In normal trypanosome cell division, the

daughter cell forms at the posterior and is the one that gets the new flagellum, as shown by series of sequential images. Here, we're looking at a snapshot of a dividing cell and judging which one is the daughter based on relative position. Some additional text has been added to the figure legend and relevant paragraph (lines 286-293, revised MS) to highlight this anomaly and clarify our interpretation.

9) L 261-262: This wording is confusing: "... the 4C nucleus of the meiotic divider undergoes a phased reduction division in contrast to the equal divisions of conventional meiosis." As you explain, the first meiotic division ($4C \rightarrow 2 \times 2C$) is normal (L 294). The first meiotic division is conventionally called the reductional division, thus "phased reduction division" is misleading. It is the second, the equational, division, which is phased, as it initially takes place in only one 2C nucleus ($2 \times 2C \rightarrow 2C + C + C$).

Sorry, our error – this is indeed misleading and has been rewritten. (line 363, "the 4C nucleus undergoes a conventional reduction division yielding two 2C nuclei").

10) L 367-369: The terms "symmetrical" and "asymmetrical divisions" should be avoided. They are rather used in the sense of equal or unequal distribution of chromosomes (e.g., sex chromosomes, B-chromosomes) or equal/unequal sized products (e.g., egg and polar bodies). Better stay with "asynchronous second divisions".

In this concluding paragraph, we already made this distinction, using "asymmetric" to refer to formation of polar body during oogenesis, and "asynchronous" to refer to the trypanosome nuclear divisions observed here.

11) L 495: Correct spelling of the author is Hörandl

Thanks, corrected.

12) It would help the reader to see stages shown in Figs 2, 4, and 6 linked to the classes A-C described in Fig. 7, maybe by referring to corresponding panels in Figs 2, 4, 6 in the chapter "Model of meiosis"

This info has been added to the text describing Figure 7 (lines 371-419 revised MS).

REVIEWERS' COMMENTS:

Reviewer #1 (Remarks to the Author):

The authors have carefully considered most comments and suggestions to proposed a very nice revised version. The addition of a PCA of morphometric parameters and the use of the YL1/2 antibody to unravel the diversity of kinetoplast profiles (although this was not as fruitful as expected) are appreciable and reinforce the analysis.

The paragraph "Model of meiosis" is possibly still hard to follow, especially for a non-specialist, but I guess simplifying it without losing the required level of precision could be challenging.

The mystery around BB-free and flagellum-free extra-kinetoplasts still remains to be unraveled in the future...

In the abstract (l.32), introduction (l.74) and result (l.142) sections, abbreviations should be defined as they occur at first.

Reviewer #2 (Remarks to the Author):

I agree with Reviewer I in that the model of gamete formation in *Trypanosoma* may be incomplete and not supported by all data. Nevertheless, the absence of 4N cells (with 4 nuclei) and the presence of 3N cells convinces me that this meiosis is indeed unusual and produces gametes in a sequential fashion. However, observations leave open the possibility that staged gamete formation is a feature of only part of the "meiotic dividers".

My concerns were satisfactorily addressed by the authors, but I have a comment on the organization of the Abstract, which may be a matter of taste. Like the main body of a paper also the Abstract should be grouped in intro, results and conclusion. In the sentence "We propose a model whereby the two nuclei resulting from Meiosis I undergo asynchronous second meiotic divisions with sequential production of haploid gametes." you summarize and interpret the main findings of the paper. However, this is weakened by two final observations that sound like an afterthought. Moreover, the phrase "these meiotic intermediates" (L33) refers to something that is not explained in the previous sentence but further up. This disrupts the context. Therefore, I would place the "We propose a model..." as the last sentence. Also note, that I have slightly changed it: "2C" is superfluous and confusing here and the "asynchronous division" should be specified as the "second meiotic divisions" for clarity.

Minor comments:

L90: The wording "In *T. brucei*, there is a HAP2 homologue ..., but its expression and function are yet to be elucidated." casts doubt on HAP2 as a marker of gametes.

L311: "While the majority of gametes expressed HAP2, 16% (35 of 220 gametes) were negative, comprising 10 1K1N and 25 2K1N gametes (Table 2). It is possible that these non-expressors were female gametes, as only male gametes express HAP2 in other organisms..." This is not a satisfactory explanation as the frequency of HAP2 non-expressing gametes would have to be close to 50%. Also, the non-HAP2 gametes do not assort with the presence of kinetoplasts.

Was it possible to measure "DNA contents" of different cell types side-by-side? Among such groups, cases should be found where fluorescence intensities are in ratios of 4:2:1 corresponding to 4C, 2C and 1C nuclei. This would provide support for the interpretation that the intermediate stages indeed contain 2C and 1C nuclei.

The graphical representation of data in Fig. 5 is unprecise. The heights of columns representing frequencies should sum up to 100. I measured only 83 in 5b (slightly better in 5a).

Response to REVIEWERS' COMMENTS:

Reviewer #1 (Remarks to the Author):

The authors have carefully considered most comments and suggestions to proposed a very nice revised version. The addition of a PCA of morphometric parameters and the use of the YL1/2 antibody to unravel the diversity of kinetoplast profiles (although this was not as fruitful as expected) are appreciable and reinforce the analysis.

The paragraph "Model of meiosis" is possibly still hard to follow, especially for a non-specialist, but I guess simplifying it without losing the required level of precision could be challenging.

The mystery around BB-free and flagellum-free extra-kinetoplasts still remains to be unraveled in the future...

In the abstract (l.32), introduction (l.74) and result (l.142) sections, abbreviations should be defined as they occur at first.

Abbreviations checked and defined. Not possible to expand GCS1 without exceeding word limit for abstract, but this is expanded in Introduction.

Reviewer #2 (Remarks to the Author):

I agree with Reviewer 1 in that the model of gamete formation in *Trypanosoma* may be incomplete and not supported by all data. Nevertheless, the absence of 4N cells (with 4 nuclei) and the presence of 3N cells convinces me that this meiosis is indeed unusual and produces gametes in a sequential fashion. However, observations leave open the possibility that staged gamete formation is a feature of only part of the "meiotic dividers".

My concerns were satisfactorily addressed by the authors, but I have a comment on the organization of the Abstract, which may be a matter of taste. Like the main body of a paper also the Abstract should be grouped in intro, results and conclusion. In the sentence "We propose a model whereby the two nuclei resulting from Meiosis I undergo asynchronous second meiotic divisions with sequential production of haploid gametes." you summarize and interpret the main findings of the paper. However, this is weakened by two final observations that sound like an afterthought. Moreover, the phrase "these meiotic intermediates" (L33) refers to something that is not explained in the previous sentence but further up. This disrupts the context. Therefore, I would place the "We propose a model..." as the last sentence. Also note, that I have slightly changed it: "2C" is superfluous and confusing here and the "asynchronous division" should be specified as the "second meiotic divisions" for clarity.

Thanks for helpful suggestions – abstract revised accordingly and also reduced to fit word limit.

Minor comments:

L90: The wording "In *T. brucei*, there is a HAP2 homologue ..., but its expression and

function are yet to be elucidated." casts doubt on HAP2 as a marker of gametes.
Qualified to "yet to be elucidated in trypanosomes".

L311: "While the majority of gametes expressed HAP2, 16% (35 of 220 gametes) were negative, comprising 10 1K1N and 25 2K1N gametes (Table 2). It is possible that these non-expressors were female gametes, as only male gametes express HAP2 in other organisms..." This is not a satisfactory explanation as the frequency of HAP2 non-expressing gametes would have to be close to 50%. Also, the non-HAP2 gametes do not assort with the presence of kinetoplasts.

This statement has been qualified "It is possible (though unlikely given the small proportion) that these non-expressors were female gametes...".

Was it possible to measure "DNA contents" of different cell types side-by-side? Among such groups, cases should be found where fluorescence intensities are in ratios of 4:2:1 corresponding to 4C, 2C and 1C nuclei. This would provide support for the interpretation that the intermediate stages indeed contain 2C and 1C nuclei.

We avoided potential inaccuracy of cell to cell comparisons and only measured DNA contents of nuclei within the same cell.

The graphical representation of data in Fig. 5 is unprecise. The heights of columns representing frequencies should sum up to 100. I measured only 83 in 5b (slightly better in 5a).

The columns represent numbers of cells with a particular nuclear ratio and therefore do not add to 100.